# Associations of biological ageing and genetic risk with incident abdominal aortic aneurysm
Chen Yao [1,5], Guochang You[1,5], Runnan Shen[2,5], Kangjie Wang[1], Yunhao Sun[1], Xiong Chen [3] &
Kai Huang [4] ✉

## Abstract

**Background** Abdominal aortic aneurysm (AAA) is a degenerative cardiovascular disorder prevalent with ageing. While accelerated biological ageing contributes to age-related diseases, its specific role in AAA risk remains unclear. This study investigates the relationships between biological ageing and risk of incident AAA and genetic predisposition to the disease.

**Methods** This retrospective study used UK Biobank data from 350,483 participants without AAA at baseline. Biological age was assessed using Klemera-Doubal Method (KDMAge) and phenotypic age (PhenoAge) algorithms. Accelerated biological ageing was determined through residual analysis against chronological age, with values above 0 indicating accelerated ageing. Cox proportional hazards models evaluated the associations of biological age accelerations with AAA risk. Polygenic risk scores were calculated to evaluate genetic predisposition to AAA. We also examined the interactions between biological age accelerations and genetic predisposition on the risk of AAA.

**Results** Here we show that participants with accelerated biological ageing have an elevated risk of AAA onset compared to those without, with hazard ratios (HRs) of 1.29 (95% CI 1.17-1.42) for KDMAge and 1.63 (95% CI 1.47-1.81) for PhenoAge. For joint associations, participants with accelerated biological ageing and high genetic risk have the highest risk of incident AAA (KDMAge: HR 2.15, 95% CI 1.81-2.54; PhenoAge: HR 2.72, 95% CI 2.26-3.28). There is a significant additive interaction between high genetic risk and accelerated biological ageing of PhenoAge.

**Conclusions** Accelerated biological ageing is significantly associated with an increased risk of AAA incidence, suggesting its potential as a focal point for risk assessment and targeted intervention development.

## Plain language summary

Abdominal aortic aneurysm (AAA) is a life-threatening cardiovascular degenerative disease, for which ageing is a non-negligible risk factor. However, the specific role of accelerated biological ageing remains unclear. We investigated this relationship in over 350,000 participants in the UK, measuring biological age using clinical biomarker-based algorithms and assessing its association with future AAA risk, alongside genetic predisposition. We found that accelerated biological ageing significantly increased the risk of developing AAA, with the highest risk observed in individuals who also had high genetic risk. These findings establish biological ageing as a significant risk factor for AAA. Assessing biological age, especially in combination with genetic risk, could help identify at-risk individuals early and guide targeted prevention strategies.

Abdominal aortic aneurysm (AAA) is a progressive and life-threatening cardiovascular degenerative disorder, most common in men over 65 years of age[1]. AAA affects over 1 million individuals in the United States, and abdominal aortic rupture is associated with 50–80% mortality, posing a growing clinical and public health challenge[2]. Besides the major modifiable risk factors like smoking, ageing is a significant and non-negligible risk factor for AAA[3]. However, defining age solely by the number of years since birth, that is, chronological age, fails to capture the nuanced complexities of ageing[4]. Therefore, it is crucial to identify the impact of biological age, as defined based on ageing biomarkers, in the development of AAA.

Recent studies have introduced measures of biological ageing that use clinical traits to depict the ageing spectrum across various organs and

[1]Division of Vascular Surgery, The First Affiliated Hospital, Sun Yat-sen University, Guangzhou, Guangdong Province, P.R. China. [2]Department of Cardiology, Sun Yat-sen Memorial Hospital, Sun Yat-sen University, Guangzhou, Guangdong Province, P.R. China. [3]Department of Urology, Sun Yat-sen Memorial Hospital, Sun Yat-sen University, Guangzhou, Guangdong Province, P.R. China. [4]Department of Cardiovascular Surgery, Sun Yat-sen Memorial Hospital, Sun Yat-sen University, Guangzhou, Guangdong Province, P.R. China. [5]These authors contributed equally: Chen Yao, Guochang You, Runnan Shen.
✉e-mail: huangk37@mail.sysu.edu.cn

systems[5,6]. Algorithms amalgamating standard clinical parameters have been highly acclaimed for their effectiveness in predicting morbidity and mortality[7,8]. To date, the Klemera-Doubal method (KDMAge) and phenotypic age (PhenoAge) provide exhaustive evaluations of an individual's biological age based on clinical chemistry biomarkers. Both methodologies have proven effective in diverse cohorts of older adults, accurately predicting disease onset, functional disability, and mortality[9,10]. However, the nexus between accelerated biological ageing and the incidence of AAA has not yet been investigated.

AAA is highly heritable, with family and twin studies estimating its heritability at around 70%[11]. Numerous loci have been identified as being associated with AAA development through genome-wide association studies (GWASs)[12,13]. Polygenic risk scores (PRSs), developed using summary statistics from GWASs, aggregate the effects of variant profiles and can effectively quantify an individual's genetic predisposition to the disease[14]. Despite the well-established role of genetics in AAA etiology, few studies have examined the interaction between genetic predisposition and accelerated biological ageing on the risk of AAA incidence.

While chronological age serves as a crude proxy for cumulative risk exposure[15], we hypothesize that individual variation in biological ageing trajectories as captured by ageing biomarkers, quantified as biological age acceleration independent of chronological age, may independently contribute to AAA development. To fill this knowledge gap, we perform a retrospective analysis of prospectively collected data examining the relationship between biological ageing and the risk of incident AAA among nearly 0.35 million participants from the UK Biobank, with a follow-up period exceeding 13 years. We scrutinize the associations between biological age accelerations of these two biological age metrics and the risk of AAA incidence. Furthermore, we investigate the joint effects and interactions of biological age accelerations and PRS in the development of AAA. Additionally, we investigate whether biological age accelerations mediated the pathogenic process of smoking, one of the most significant environmental risk factors for AAA. Our findings demonstrate that accelerated biological ageing is significantly associated with an increased risk of incident AAA. Besides, individuals who exhibit both accelerated biological ageing and a high genetic predisposition to AAA have the highest risk of developing the disease. A significant additive interaction exists between high genetic risk and accelerated biological ageing of PhenoAge. Moreover, accelerated biological ageing partially mediates the detrimental effect of smoking on AAA incidence.

## Methods

### Study design and participants

The UK Biobank is a population-based prospective study involving around 0.5 million participants aged 37 to 73 years, recruited from 22 centers across the UK between 2006 and 2010, with several follow-up evaluations conducted thereafter[16]. At recruitment, participants were required to complete touchscreen questionnaires, verbal interviews, and physical measurements for collecting comprehensive exposures and health status. In addition, biological samples were also provided by participants for use in a range of assay types. Further details regarding the UK Biobank protocol can be assessed through the website (http://www.ukbiobank.ac.uk/). Written informed consent was obtained from all participants, and the study protocol was reviewed and approved by the North West Multi-Centre Research Ethics Committee. Our research was conducted with the UK Biobank Resource under application number 100739.

In this study, we excluded participants with missing values in any biomarker required for the calculation of the two biological age measures ($n = 147,247$). Subsequently, participants with missing genetic data were excluded ($n = 2449$). Furthermore, participants were excluded from the study if they possessed a baseline history of AAA, self-reported aortic aneurysm, aortic dissection, or cerebral aneurysm, and those who were hospitalized with a vascular disease diagnosis either prior to or within 30 days following the UK Biobank assessment date ($n = 1991$)[17]. Following the exclusion criteria, a final cohort of 350,483 participants was included (Fig. 1).

### Assessment of biological ageing

The KDMAge and PhenoAge algorithms were performed to assess biological ageing using blood-chemistry-derived biomarkers. KDMAge was developed through forced expiratory volume in one second, systolic blood pressure, and seven blood-chemistry metrics (albumin, alkaline phosphatase, blood urea nitrogen, creatinine, C-reactive protein, glycated hemoglobin, and total cholesterol). The equation is:

$$\text{KDMAge} = \frac{\sum_{i=1}^{n} (x_i - q_i)\frac{k_i}{s_i^2} + \frac{CA}{s_{BA}^2}}{\sum_{i=1}^{n} \left(\frac{k_i}{s_i}\right)^2 + \frac{1}{s_{BA}^2}}$$

Where $x_i$ represents the value of each biomarker for an individual; $k_i$, $q_i$, and $s_i$ are the intercept, slope, and root mean square error, respectively, estimated from the regression of chronological age (CA) on each biomarker separately for men and women; $s_{BA}$ is the square root of the variance in CA explained by the set of biomarkers; and $n = 9$, which is the total number of biomarkers used in this study.

PhenoAge was derived with nine multisystem blood-chemistry biomarkers, including four that coincide with KDMAge (albumin, alkaline phosphatase, creatinine, C-reactive protein, glucose, lymphocyte proportion, mean cell volume, red cell distribution width, and white blood cell count)[18,19]. The formula is:

$$\text{PhenoAge} = 141.50 + \frac{\ln[-0.00553 \times \ln(1 - mortality\ risk)]}{0.09165}$$

$$mortality\ risk = 1 - \exp\left(\frac{-1.51714 \times \exp(xb)}{0.0076927}\right)$$

$xb = -19.907 - 0.0336 \times$ albumin $+ 0.0095 \times$ creatinine $+ 0.1953 \times$ glucose $+ 0.0954 \times \ln$ (C-reactive protein) $- 0.0120 \times$ lymphocyte proportion $+ 0.0268 \times$ mean cell volume $+ 0.3306 \times$ red blood cell distribution width $+ 0.00188 \times$ alkaline phosphatase $+ 0.0554 \times$ white blood cell count eq.$+ 0.0804 \times$ chronological age.

Each biomarker was winsorized by setting the bottom and top 1% of values to their respective 1st and 99th percentiles to adjust for skewed distributions[20]. Furthermore, to evaluate variations in biological ageing across participants, we calculated residual values derived from the regression of biological age relative to baseline chronological age, which were considered as biological age accelerations. Subsequently, accelerated biological ageing was characterized by a KDMAge or PhenoAge acceleration value above zero, while non-accelerated biological ageing was identified with the value of zero or less. The selected biomarkers and algorithms, along with their corresponding R code, are provided in the "BioAge" R package, accessible at https://github.com/dayoonkwon/BioAge, and detailed in the associated publications[21].

### Polygenic risk score calculation

For the construction of the PRS for AAA, imputed genotype data sourced from the UK Biobank was analyzed. The comprehensive genotyping process, imputation techniques, and quality control measures applied within the UK Biobank have been documented elsewhere[22]. Based on findings from a high-quality cohort study[23], we selected 31 independent single-nucleotide polymorphisms (SNPs) associated with AAA, as identified from two large-scale GWASs[12,13]. These SNPs, devoid of linkage disequilibrium ($r^2 < 0.05$ or spaced over 1000 kb apart), have been identified as significantly associated with the incidence of AAA in individuals of European ancestry, featuring a minor allele frequency greater than 0.05 and a $P$ value less than $1 \times 10^{-5}$. Additional information about these SNPs can be found in Supplementary Data 1. We generated the PRS using a weighted approach previously described in the literature[24]. A higher PRS reveals a higher genetic predisposition to developing AAA. In our analysis, participants were categorized into low, intermediate, or high genetic risk groups based on PRS tertiles. The

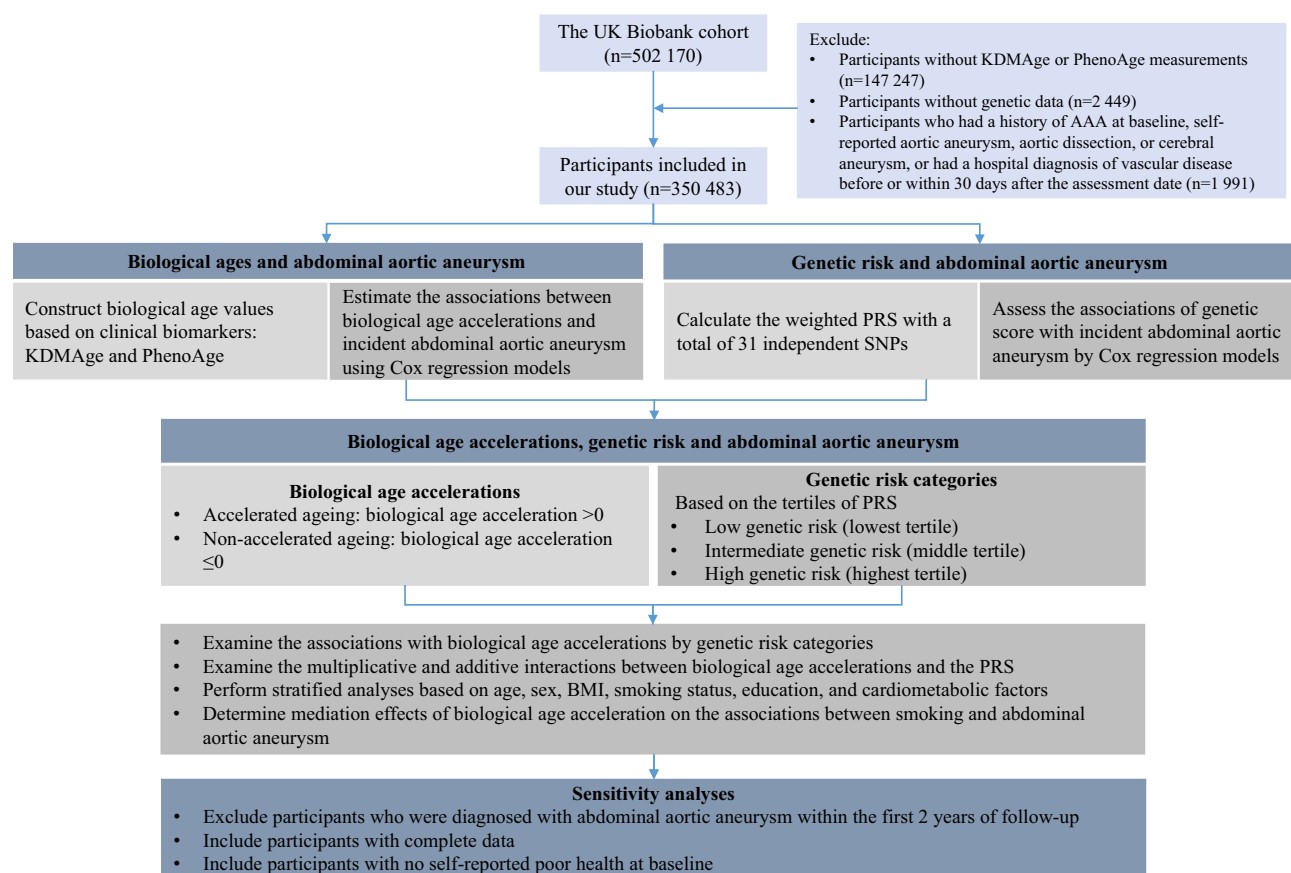

**Fig. 1 | Study design and flowchart.** AAA abdominal aortic aneurysm, BMI body mass index, KDMAge biological age calculated by the Klemera-Doubal method, PhenoAge phenotypic age, PRS polygenic risk score, SNP single-nucleotide polymorphism.

weighted PRS was calculated as follows:

$$\text{Weighted PRS} = \sum_{i=1}^{n} \beta_i SNP_i \times \frac{N}{Sum(\beta)}$$

Here, $\beta$ denotes the per-allele log odds ratios for AAA associated with each $SNP$, derived from previous GWAS studies. $SNP$ was coded as 0, 1, or 2, depending on the number of risk alleles. The variable $N$ represents the total number of selected $SNPs$.

#### Outcome ascertainment

In our study, AAA diagnoses were ascertained through hospital inpatient, death, and procedure records in the UK Biobank, defined according to prior literature utilizing the UK Biobank[17,23,25]. We identified relevant diagnoses using the International Classification of Diseases 10th revision (ICD-10), for the first hospital inpatient diagnosis or death of AAA, and the Office of Population, Censuses and Surveys: Classification of Interventions and Procedures (OPCS-4) codes for AAA-related surgical procedures. All diagnosis codes used in our study are provided in Table S1. Participants were followed up from the enrollment until their first AAA diagnosis (incident AAA), death, lost to follow-up date, or to the last date of hospital admission from the respective database (October 31, 2022, in England; August 31, 2022, in Scotland; and May 31, 2022, in Wales).

#### Covariates

Potential covariates were assessed using multiple data sources, comprising baseline questionnaires, physical examinations, and medical history. The following variables are considered potential confounders: chronological age (years), sex (male or female), ethnicity (Asian, Black, White, or Others),

body mass index (BMI, kg/m²), education (degree-level education or non-college education), employment (employment, retired, or unemployment), Townsend Deprivation Index, smoking status (never smoker or current or former smoker), pack-years of smoking, alcohol consumption frequency (<3 times per week, ≥3 times per week, or never), healthy diet score[26] (range from 0 to 10 points, Table S2), physical activity[27] (never, low, medium, or high activity), self-reported health (excellent, fair, good, or poor), use of blood pressure medication (yes or no), use of insulin medication (yes or no), use of cholesterol-lowering medication (yes or no).

The variables used in this study and the corresponding field ID in the UK Biobank are shown in Table S3. For categorical covariates, missing values were assigned to an "unknown" category, whereas missing values for continuous covariates were imputed using the conditional mean, stratified by sex.

#### Statistics and reproducibility

Descriptive statistics were presented using median (interquartile range, IQR) or mean (standard deviation, SD) for continuous measures, and as frequencies (percentage) for categorical parameters. To assess the associations of biological age accelerations with the risk of incident AAA, Cox proportional hazards regression models were implemented to estimate hazard ratios (HRs) and 95% confidence interval with adjustment for age, sex, ethnicity, body mass index, education, employment, Townsend Deprivation Index, smoking status, pack-years of smoking, alcohol consumption frequency, healthy diet score, physical activity, self-reported health, blood pressure medication, insulin medication, cholesterol-lowering medication. For the analysis concerning genetic predisposition, this study included only participants of European ancestry, comprising a total of 331,888 participants. Additionally, the models were also adjusted for the

genotyping batch and the first ten genetic principal components. Using the Schoenfeld residuals method, no violations of the proportional hazard assumption were detected. We calculated the trend $P$ value among increasing exposure groups utilizing the integer numbers ranging from 1 to 4. Dose-response relationships between biological age accelerations and incident AAA were examined by a restricted cubic spline model, with three knots specified at the 25th, 50th, and 75th centiles.

To evaluate whether genetic factors impact the association between accelerated biological ageing and AAA risk on a multiplicative scale, we introduced an interaction term into the Cox models and computed the $P$ value for interaction. We assessed additive interactions through calculations of the relative excess risk (RERI) and the attributable proportion (AP), derived from the product term's coefficients[28]. The 95% confidence intervals for RERI and AP were established using 1000 bootstrap samples, with 0 included in these intervals indicating no additive interaction. Moreover, we conducted a mediation analysis with the "mediation" R package, incorporating biological age acceleration as a mediator to investigate to what extent the adverse impact of smoking on AAA onset could be mediated by biological age accelerations[29]. We also carried out subgroup analyses grouped by age, sex, BMI, smoking status, education, and cardiometabolic factors, including hypertension, diabetes, and dyslipidaemia, to identify potentially susceptible populations.

Several sensitivity analyses were conducted to ensure the robustness of our results with respect to biological ageing measures. First, a landmark analysis was implemented by moving the start of follow-up to 2 years after baseline visit and by excluding individuals with diagnosed AAA to account for potential reverse causality. Second, we performed a complete-case analysis, using only available data and excluding individuals with missing information on any covariates. Third, we restricted our sample to participants who did not report poor health at baseline. Finally, we reanalyzed the relationship between biological age accelerations and AAA risk with the Fine and Gray competing risk model, which adjusts for the competing risk of death. A two-tailed $P$ value of less than 0.05 suggested statistical significance. All analyses were performed in R software (version 4.2.3).

## Results

### Population characteristics

Baseline characteristics of the participants based on AAA status are provided in Supplementary Data 2, 3. Of the 350,483 eligible participants, the mean age at baseline was 56.4 (±8.1) years, 189,698 (54.1%) were female, and 155,832 (44.4%) were current or former smokers. Over a median (IQR) follow-up of 13.68 (12.96–14.31) years, 1886 incident AAA cases were recorded. Participants with AAA generally show older chronological ages, a higher prevalence of males, increased rates of current or former smoking, lower educational attainment, poorer employment status, and a tendency to take blood pressure and cholesterol-lowering medications, along with advanced biological ages. Additionally, 171,573 (49.0%) participants of the whole population exhibited accelerated ageing as determined by the KDMAge measure, and 162,492 (46.4%) participants were identified as experiencing accelerated ageing according to the PhenoAge metric. Among these, 97,933 (27.9%) individuals met the criteria for accelerated ageing on both the KDMAge and PhenoAge measures. Biological age acceleration distributions are shown in Table S4 and Fig. S1. The mean (SD) of biological age acceleration were 0.004 (2.386) for KDMAge, and −0.019 (5.045) for PhenoAge.

### Accelerated biological ageing and incident AAA

Participants with incident AAA showed increased biological age acceleration relative to those who did not develop AAA (Fig. S2). In our analyses, we found that for every 2.386-unit increase in KDMAge acceleration and 5.045-unit increase in PhenoAge acceleration, the risk of incident AAA increased by 6% (Table 1, see also Supplementary Data 4 for full Cox model estimates). Similar associations were also detected upon fitting the model with biological age acceleration categories from quartile 1 to quartile 4, yielding a significant trend ($P < 0.001$). Restricted cubic spline analyses revealed a

**Table 1 | Association between biological age acceleration and incident abdominal aortic aneurysm risk**

| | Hazard ratio (95% CI) | P value | P for trend |
|---|---|---|---|
| **KDMAge acceleration** | | | |
| Quartile 1 | Ref. | | <0.001 |
| Quartile 2 | 1.23 (1.07, 1.42) | 0.004 | |
| Quartile 3 | 1.30 (1.13, 1.50) | <0.001 | |
| Quartile 4 | 1.60 (1.39, 1.83) | <0.001 | |
| Continuous, per SD increase | 1.06 (1.04, 1.09) | <0.001 | |
| Non-accelerated ageing[*] | Ref. | | |
| Accelerated ageing[†] | 1.29 (1.17, 1.42) | <0.001 | |
| **PhenoAge acceleration** | | | |
| Quartile 1 | Ref. | | <0.001 |
| Quartile 2 | 1.32 (1.11, 1.58) | 0.002 | |
| Quartile 3 | 1.56 (1.32, 1.85) | <0.001 | |
| Quartile 4 | 2.38 (2.02, 2.81) | <0.001 | |
| Continuous, per SD increase | 1.06 (1.05, 1.07) | <0.001 | |
| Non-accelerated ageing[*] | Ref. | | |
| Accelerated ageing[†] | 1.63 (1.47, 1.81) | <0.001 | |

Model was adjusted for age, sex, ethnicity, body mass index, education, employment, Townsend Deprivation Index, smoking status, pack-years of smoking, alcohol consumption frequency, healthy diet score, physical activity, self-reported health, blood pressure medication, insulin medication, and cholesterol-lowering medication. Two-sided $P$ values were calculated using the Wald test based on the Cox proportional hazards regression models. No multiple comparison adjustment was applied. *CI* confidence interval, *KDMAge* biological age calculated by the Klemera-Doubal method, *PhenoAge* phenotypic age, *SD* standard deviation.

[*]Non-accelerated ageing indicates that the residual of the regression of KDMAge or PhenoAge based on chronological age is ≤0.

[†]Accelerated ageing indicates that the residual of the regression of KDMAge or PhenoAge based on chronological age is >0.

linear dose-response association between KDMAge acceleration ($P$ for non-linearity: 0.390) and PhenoAge acceleration ($P$ for non-linearity: 0.750) in relation to the incidence of AAA, without evidence of a minimal or maximal thresholds (Fig. 2). Participants with accelerated biological ageing had a significantly increased risk of developing AAA compared to those with no accelerated ageing, as indicated by both KDMAge (HR 1.29 [95% CI 1.17–1.42]; $P < 0.001$) and PhenoAge (HR 1.63 [95% CI 1.47–1.81]; $P < 0.001$; Table 1). The cumulative incidence of AAA was higher among participants with accelerated biological ageing compared to those without, as depicted in Fig. S3. The associations between accelerated biological ageing, as quantified by KDMAge and PhenoAge, and the risk of AAA were stronger in participants with a BMI under 25 kg/m$^2$ compared to those with higher BMIs (Table S5). Also, the impact of accelerated biological ageing, as assessed by PhenoAge, on AAA risk was greater in current or former smokers than in individuals who never smoked. No significant interactions were found in additional stratified analyses involving cardiometabolic factors.

### Stratified and joint analysis

For PRS, participants with incident AAA had a significantly higher distribution compared to those without (Fig. S4). Additionally, the dose-response analysis indicated a positive linear relationship between PRS and incident AAA ($P$ for non-linearity: 0.550). Participants with intermediate or high genetic risk had a 33% (17%, 50%) or 69% (51%, 90%) increased risk of AAA onset relative to those at low genetic risk (Table S6). Stratified analyses by genetic risk categories indicated that elevated biological age acceleration

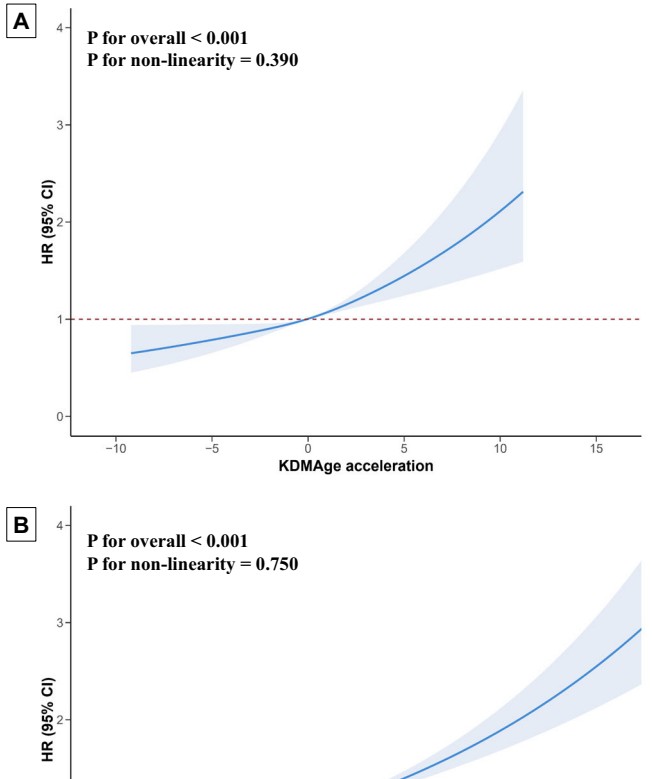

**Fig. 2 | The dose-response association of biological age accelerations with the risk of incident abdominal aortic aneurysm.** Restricted cubic spline models for the association between **A** KDMAge acceleration and **B** PhenoAge acceleration and the risk of AAA incidence among participants in the UK Biobank ($N = 350,483$). Hazard ratios (solid lines) and 95% confidence intervals (shaded areas) were adjusted for age, sex, ethnicity, body mass index, education, employment, Townsend Deprivation Index, smoking status, pack-years of smoking, alcohol consumption frequency, healthy diet score, physical activity, self-reported health, blood pressure medication, insulin medication, and cholesterol-lowering medication. The blue solid curve is the point estimate of the hazard ratio (center) from the restricted cubic spline (RCS) Cox model; the shaded band denotes the model-based 95% confidence interval (error band) around the estimated HR. Overall and non-linear association *P* values are calculated from Wald tests of the spline terms (anova on the RCS model), two-sided. The reference value (hazard ratio = 1; dotted horizontal line) was set by the median exposure variable (−0.0608 for KDMAge acceleration, and −0.4354 for PhenoAge acceleration). CI indicates confidence interval, HR hazard ratio, KDMAge biological age calculated by the Klemera-Doubal method, *N* number of participants included in analysis, PhenoAge phenotypic age.

remained associated with a higher AAA risk across all genetic groups, notably in the high genetic risk category (Table 2). Participants with a high genetic risk for AAA showed a significantly increased risk of developing the disease when experiencing accelerated biological ageing (KDMAge: HR 1.28 [1.10–1.50]; PhenoAge: HR 1.67 [1.41–1.97]). Additionally, we assessed the joint effects of biological age accelerations and genetic predisposition on the risk of AAA onset (Fig. 3). Compared with participants with low genetic risk and non-accelerated ageing, participants with accelerated biological ageing and high genetic risk had the highest risk of incident AAA (KDMAge: HR 2.15 [1.81–2.54], *P* < 0.001; PhenoAge: HR 2.72 [2.26–3.28], *P* < 0.001). The most significant harmful impact was found in participants with both high

genetic risk and the highest quartile (quartile 4) of biological age acceleration, relative to those with low genetic risk and the lowest quartile (KDMAge: HR 2.53 [2.00–3.21], *P* < 0.001; PhenoAge: HR 3.72 [2.77–4.98], *P* < 0.001). The results of the interaction analysis indicated a significant additive interaction between high genetic risk and accelerated biological ageing, as assessed by PhenoAge (Table 3). For individuals with high genetic risk and accelerated biological ageing of PhenoAge, the RERIs were 0.52 (0.15–0.88), and the APs were 0.19 (0.05–0.32). However, no significant multiplicative interaction between genetic risk categories and accelerated biological ageing was observed.

## Mediation effects of smoking
We identified a significant association between exposure to smoking and increased risks of incident AAA (Table S7). Besides, our study found that exposure to smoking significantly accelerates biological ageing (Table S8). The mediation analysis revealed that PhenoAge acceleration served as a mediator between smoking and the occurrence of AAA (IE = 7.42e-06, DE = 7.11e-05, proportion of mediation = 9.45%, *P* < 0.001), with KDMAge acceleration also playing a significant mediating role (IE = 2.58e-06, DE = 7.46e-05, proportion of mediation = 3.34%, *P* < 0.001) (Fig. 4). Despite the significant mediation effects observed, the mediation proportions for KDMAge acceleration were relatively small (Table S9).

## Sensitivity analysis
We conducted four further sensitivity analyses. The pattern of associations between biological age accelerations and the risk of incident AAA remained substantially robust when we removed participants diagnosed with AAA within the first 2 years of follow-up (Table S10), excluded participants with missing covariates (Table S11), and conducted analyses among participants who did not self-report poor health status at baseline (Table S12). Furthermore, similar results as for the original analysis were observed in the Fine and Gray competing risk models (Table S13).

## Discussion
To the best of our knowledge, this is the first population-based retrospective cohort study of prospectively collected data that evaluates the association between accelerated biological ageing and the risk of AAA incidence. The main findings were that accelerated biological ageing was significantly associated with an increased risk of AAA incidence. When examining the joint associations of biological ageing and genetic risk with AAA, participants with both accelerated biological ageing and a high genetic predisposition to AAA exhibited the highest risk of incident AAA. Besides, there was a significant additive interaction between high genetic risk and accelerated biological ageing of PhenoAge. Furthermore, we also found that accelerated ageing partially mediated the adverse effects of smoking on AAA. Collectively, these findings provide future directions for AAA risk assessment and encourage proactive interventions within these demographics.

Recent research more frequently identifies biological ageing as a significant precursor to cardiovascular events[30,31], and cellular senescence has been found to correlate with the pathogenesis of AAA[32,33]. Atturu et al[34] compellingly identified leukocyte telomere length as an independent predictor of AAA susceptibility in a case-control study, which was consistently verified in the subsequent studies[35,36]. The telomere length more directly reflects cellular processes but provides fewer insights into organ-specific or whole-body ageing[37]. Thus, its application as an indicator for biological ageing may not effectively capture the association with AAA. Furthermore, composite biomarkers might better capture the multifaceted nature of biological ageing compared to individual biomarkers[38]. KDMAge and PhenoAge provide a comprehensive measurement of biological age that integrates several aspects, involving inflammation, immunity, metabolism, and organ homeostasis, enabling the effective identification of people who exhibit physiological ageing beyond their chronological age[39]. The existing literature is still deficient in exploring the interplay between comprehensive biological age markers and AAA. Our study contributes to this area by

**Table 2 | Association between biological age acceleration and incident abdominal aortic aneurysm risk, stratified by polygenic risk status**

| | Low genetic risk | | Intermediate genetic risk | | High genetic risk | |
|---|---|---|---|---|---|---|
| | HRs (95% CI) | P value | HRs (95% CI) | P value | HRs (95% CI) | P value |
| KDMAge acceleration | | <0.001* | | 0.004* | | <0.001* |
| Quartile 1 | Ref. | | Ref. | | Ref. | |
| Quartile 2 | 1.24 (0.94, 1.64) | 0.123 | 1.12 (0.88, 1.43) | 0.364 | 1.37 (1.09, 1.71) | 0.007 |
| Quartile 3 | 1.29 (0.97, 1.70) | 0.075 | 1.21 (0.95, 1.54) | 0.126 | 1.42 (1.14, 1.78) | 0.002 |
| Quartile 4 | 1.65 (1.26, 2.17) | <0.001 | 1.40 (1.11, 1.78) | 0.005 | 1.63 (1.31, 2.03) | <0.001 |
| Continuous, per SD increase | 1.06 (1.02, 1.10) | 0.002 | 1.05 (1.02, 1.09) | 0.003 | 1.06 (1.03, 1.09) | <0.001 |
| Non-accelerated ageing† | Ref. | | Ref. | | Ref. | |
| Accelerated ageing‡ | 1.31 (1.08, 1.59) | 0.007 | 1.22 (1.03, 1.44) | 0.024 | 1.28 (1.10, 1.50) | 0.001 |
| PhenoAge acceleration | | <0.001* | | <0.001* | | <0.001* |
| Quartile 1 | Ref. | | Ref. | | Ref. | |
| Quartile 2 | 1.13 (0.80, 1.61) | 0.487 | 1.53 (1.12, 2.09) | 0.007 | 1.29 (0.97, 1.70) | 0.079 |
| Quartile 3 | 1.59 (1.15, 2.19) | 0.005 | 1.60 (1.18, 2.16) | 0.003 | 1.52 (1.16, 1.99) | 0.003 |
| Quartile 4 | 2.09 (1.52, 2.88) | <0.001 | 2.47 (1.85, 3.31) | <0.001 | 2.40 (1.85, 3.11) | <0.001 |
| Continuous, per SD increase | 1.06 (1.04, 1.08) | <0.001 | 1.06 (1.05, 1.08) | <0.001 | 1.07 (1.05, 1.08) | <0.001 |
| Non-accelerated ageing† | Ref. | | Ref. | | Ref. | |
| Accelerated ageing‡ | 1.61 (1.31, 1.97) | <0.001 | 1.55 (1.30, 1.86) | <0.001 | 1.67 (1.41, 1.97) | <0.001 |

Model was adjusted for age, sex, ethnicity, body mass index, education, employment, Townsend Deprivation Index, smoking status, pack-years of smoking, alcohol consumption frequency, healthy diet score, physical activity, self-reported health, blood pressure medication, insulin medication, cholesterol-lowering medication, genotyping batch, and the first ten genetic principal components. Two-sided P values were calculated using the Wald test based on the Cox proportional hazards regression models. No multiple comparison adjustment was applied.

CI confidence interval, HR hazard ratio, KDMAge biological age calculated by the Klemera-Doubal method, PhenoAge phenotypic age, SD standard deviation.

*These data were P value for the overall trend.

†Non-accelerated ageing indicates that the residual of the regression of KDMAge or PhenoAge based on chronological age is ≤0.

‡Accelerated ageing indicates that the residual of the regression of KDMAge or PhenoAge based on chronological age is >0.

showcasing the relationships among KDMAge, PhenoAge, and incident AAA. Nevertheless, additional investigations are required to elucidate the causal association between biological ageing and AAA.

As noted in previous investigations, ageing represents a complex interplay of biological processes, with different biological age predictors illuminating distinct dimensions of ageing[40]. Furthermore, a comparison across different biological age measures is beneficial, considering the lack of an established gold standard for computing biological age using clinical biomarkers. More specifically, KDMAge generally reflects the integrity and functional capacity of systems, whereas PhenoAge is employed to assess mortality risk[41]. In this study, for KDMAge acceleration, participants with accelerated biological ageing faced a greater risk of AAA compared to those without accelerated ageing, whereas these associations of PhenoAge acceleration were more pronounced. Although systolic blood pressure, a well-established risk factor for AAA[42], is incorporated into KDMAge, PhenoAge acceleration demonstrated better risk stratification performance. This discrepancy could be attributed to PhenoAge being designed to predict all-cause mortality using a broader panel of multisystem biomarkers alongside chronological age, enabling it to better capture a range of physiological decline related to inflammation, immune function, and metabolic dysregulation[5,43]. These processes are collectively relevant to the multisystem impairments involved in AAA pathogenesis[11,44]. Furthermore, although current guidelines regard the rupture risk of small AAAs (<55 mm in men, <50 mm in women) as negligible, progression into larger aneurysms remains a significant contributor to rupture and even mortality[45,46]. Given that PhenoAge is intrinsically calibrated to mortality risk, it may be better suited to reflect the systemic deterioration related to AAA development. We additionally observed the significant additive interaction between high genetic risk and accelerated biological ageing of PhenoAge. This finding emphasizes the need to prioritize anti-ageing strategies for those genetically predisposed to AAA, as these initiatives may make important public health benefits. While smoking cessation remains the cornerstone of AAA risk

reduction, future efforts could benefit from incorporating a broader range of evidence-based anti-ageing strategies. These may include regular physical activity, dietary modifications, restorative sleep, and potential pharmacological agents targeting fundamental ageing processes[47,48]. Although these approaches are not yet specifically approved for AAA prevention, they show promise in mitigating ageing and warrant further exploration in populations with accelerated ageing.

It should be noted that while KDMAge and PhenoAge incorporate biomarkers such as C-reactive protein and systolic blood pressure, which are established risk factors for AAA[49,50], these algorithms are designed to capture integrative, systemic ageing processes rather than functioning merely as collections of discrete risk factors[51]. Further, our use of age acceleration residuals inherently adjusts for chronological age, thereby capturing the variance in biological age attributable to divergent ageing trajectories[52]. This approach allows us to test whether the ageing-related physiological dysregulation captured by these algorithms is associated with AAA beyond the contribution of individual biomarkers. Additionally, the association between biological age acceleration and AAA incidence persisted even after rigorous adjustment for a set of lifestyle and clinical covariates. This suggests that the predictive utility of these measures likely stems from their ability to capture physiological dysregulation, including feedforward pathways like vascular dysfunction and chronic inflammation, which underlie both ageing and AAA pathogenesis[31,53–55]. Further studies are needed to elucidate whether these biomarkers contribute to AAA development through specific ageing pathways.

The stronger association between accelerated biological ageing and AAA risk observed in participants with BMI ≤25 kg/m² may reflect increased susceptibility due to low lean mass, or unintentional weight loss—conditions often associated with frailty and systemic physiological decline, especially while considering the aged population[56,57]. In these individuals, accelerated biological ageing likely coincides with heightened inflammatory states and increased cellular damage, which may synergistically increase

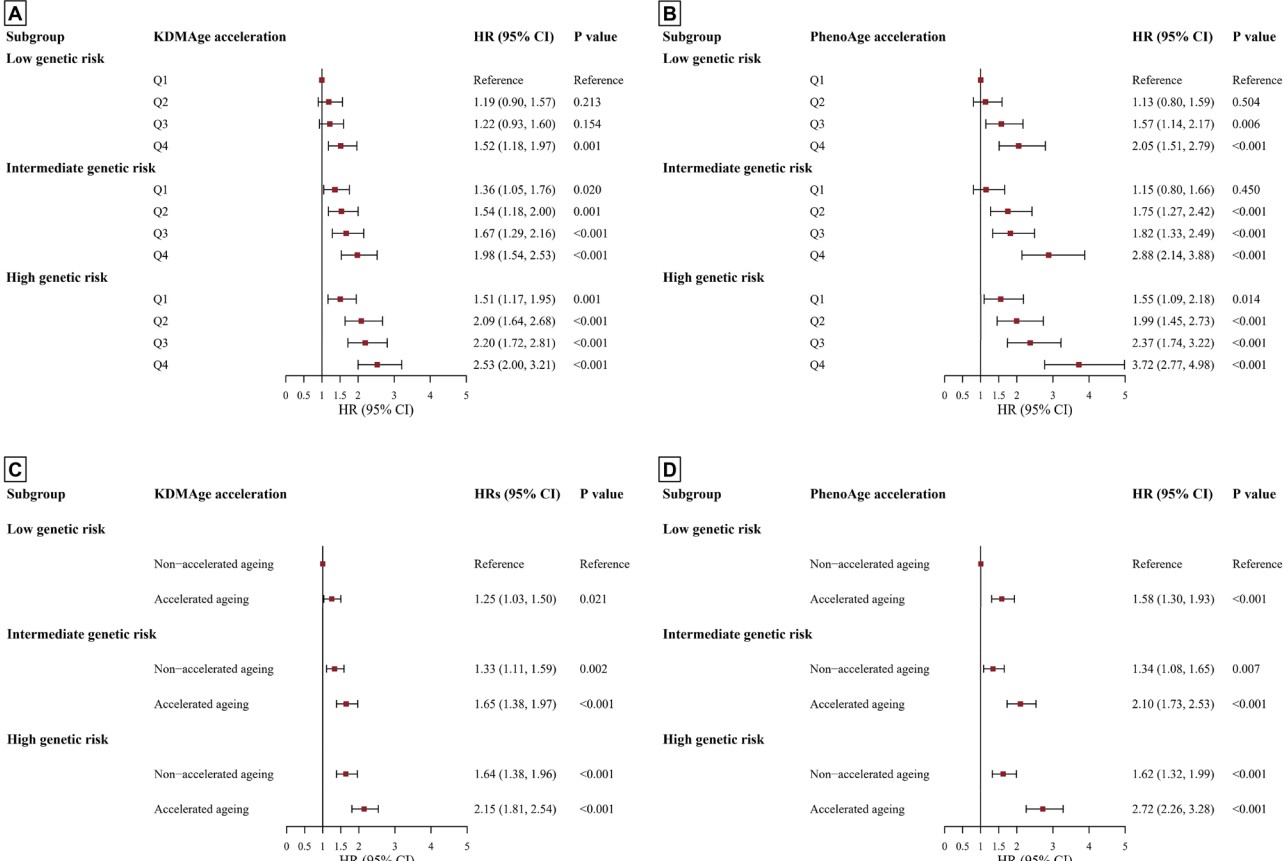

**Fig. 3 | The joint association of biological age accelerations and PRS with the risk of incident abdominal aortic aneurysm.** Estimated effects by **A** population quartiles for KDMAge, **B** PhenoAge and in the **C** population overall for KDMAge, **D** PhenoAge (N = 331,888). Hazard ratios and 95% confidence intervals were calculated using the Cox proportional hazard model adjusted for age, sex, ethnicity, body mass index, education, employment, Townsend Deprivation Index, smoking status, pack-years of smoking, alcohol consumption frequency, healthy diet score, physical activity, self-reported health, blood pressure medication, insulin medication, cholesterol-lowering medication, genotyping batch, and the first ten genetic principal components. Non-accelerated ageing indicates that the residual of the regression of KDMAge or PhenoAge based on chronological age is ≤0. Accelerated ageing indicates that the residual of the regression of KDMAge or PhenoAge based on chronological age is >0. If the 95% confidence interval overlaps with the dashed line, the results is not statistically significant. Center for the error bars indicate hazard ratios estimated by maximizing the Cox partial likelihood derived from the Cox proportional hazard models; the error bars indicate 95% confidence intervals. Two-sided P values were calculated using the Wald test. CI confidence interval, HR hazard ratio, KDMAge biological age calculated by the Klemera-Doubal method, N number of participants included in analysis, PhenoAge phenotypic age, Q quartile.

## Table 3 | Additive and multiplicative interactions between biological age accelerations and polygenic risk score on the risk of incident abdominal aortic aneurysm

| | Additive interaction | Multiplicative interaction | |
|---|---|---|---|
| | RERI (95% CI)* | AP (95% CI)* | P for interaction |
| KDMAge, accelerated ageing | | | 0.879 |
| Intermediate PRS | 0.08 (−0.24, 0.41) | 0.05 (−0.16, 0.25) | |
| High PRS | 0.26 (−0.09, 0.58) | 0.12 (−0.04, 0.27) | |
| PhenoAge, accelerated ageing | | | 0.819 |
| Intermediate PRS | 0.18 (−0.22, 0.52) | 0.09 (−0.10, 0.25) | |
| High PRS | 0.52 (0.15, 0.88) | 0.19 (0.05, 0.32) | |

If 0 is outside the CIs of RERIs and APs, it indicates that there is a significant additive interaction. Two-sided likelihood ratio tests comparing Cox proportional hazards models with and without the interaction term were used to calculate P value for the multiplicative interaction. No adjustment for multiple comparisons was applied. Model was adjusted for age, sex, ethnicity, body mass index, education, employment, Townsend Deprivation Index, smoking status, pack-years of smoking, alcohol consumption frequency, healthy diet score, physical activity, self-reported health, blood pressure medication, insulin medication, cholesterol-lowering medication, genotyping batch, and the first ten genetic principal components.
AP attributable proportion, CI confidence interval, KDMAge biological age calculated by the Klemera-Doubal method, PhenoAge phenotypic age, RERI relative excess risk.
*To estimate the RERI and AP, the non-accelerated ageing (biological age acceleration ≤0) individuals in the low PRS group was set as reference.

AAA susceptibility[58,59]. In contrast, a moderately elevated BMI may indicate preserved metabolic reserve or muscle mass, potentially mitigating the detrimental effects of biological ageing and thus attenuating the association of biological ageing with AAA risk[60,61]. This pattern aligns with the "obesity paradox" previously observed in older and frail populations, where mild adiposity has been linked to better survival, possibly owing to greater energy reserves and reduced catabolism[62]. Furthermore, reverse causality may play a role, as underlying health impairments could simultaneously drive both

**Fig. 4 | The mediation effect of biological age accelerations on the associations between pack-years of smoking and incident abdominal aortic aneurysm. A** KDMAge acceleration and **B** PhenoAge acceleration mediated the association between pack-years of smoking and abdominal aortic aneurysm. Model was adjusted for age, sex, ethnicity, body mass index, education, employment, Townsend Deprivation Index, smoking status, pack-years of smoking, alcohol consumption frequency, healthy diet score, physical activity, self-reported health, blood pressure medication, insulin medication, and cholesterol-lowering medication. *P* values were assessed using a nonparametric percentile bootstrap method with 1000 simulations (two-sided test). No multiple comparison adjustment was applied. DE direct effect, IE indirect effect.

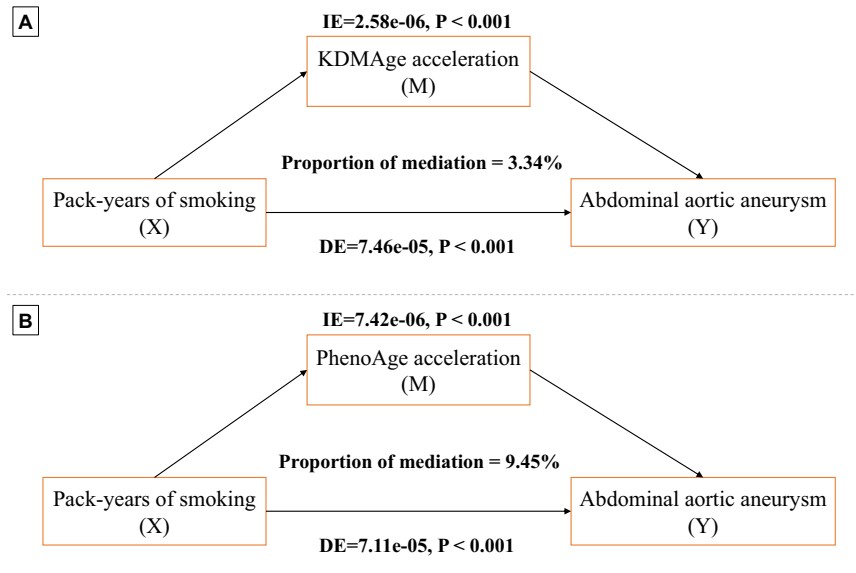

accelerated biological ageing and weight loss, or the limitations of the BMI index itself, thereby exaggerating this association observed in normal-weight individuals[63,64]. Future studies would benefit from incorporating multiple measures of adiposity alongside BMI to better clarify the roles of adiposity and biological ageing in AAA pathogenesis.

In our study, we found through mediation analysis that biological age accelerations played a mediating role between smoking and AAA. We further demonstrated that PhenoAge acceleration likely mediates ~10% of the association between smoking and AAA, while about 3% of the association is mediated by KDMAge acceleration. The potential mechanisms of accelerated biological ageing triggered by tobacco combustion may be involved in oxidative stress, mitochondrial dysfunction, and inflammation, leading to vascular remodeling and metabolism alterations[65]. As there is currently no drug therapy that has convincingly limited AAA growth in randomized controlled trials[45], efforts to alleviate the burden of AAA like smoking cessation and early non-invasive detection, may be effective. Our findings demonstrated that biological age, such as KDMAge and PhenoAge, may act as potential predictive biomarkers for AAA, which paves the way for developing non-invasive diagnostic tools for AAA detection and risk stratification.

Several strengths of our analysis deserve to be mentioned. First, it is a retrospective cohort study of prospectively collected data from the UK Biobank, characterized by a substantial sample size and an extended follow-up period, which augments the evidence regarding the association between comprehensive biomarker-based biological ageing and the risk of AAA onset. Second, our study firstly utilizes a joint analysis to explore how biological ageing and genetic risk contribute to incident AAA in gene-environment interaction contexts, potentially enhancing early interventions and accurate disease assessment. However, this study also has some limitations. First, our observational study, based on prospectively collected data, reveals an association between accelerated biological ageing and increased AAA risk, but causality is not definitively established. Future Mendelian randomization studies using robust genetic instruments derived from large-scale GWAS data may further elucidate the causal nature of their associations. Second, since the UK Biobank lacks information regarding aneurysm sizes, we could not factor these details into our analysis. Third, our diagnosis of outcome events relies on hospital inpatient and death records, possibly missing latent cases exhibiting less pronounced clinical symptoms. Furthermore, despite the existence of the UK's national AAA screening program[66], some cases may still arise from incidental detection during other imaging, potentially introducing selection bias towards individuals with higher comorbidity. Fourth, our participants were predominantly (~95%) composed of middle-aged or older White adults, which may restrict the

generalizability of our findings to more diverse populations[5]. Fifth, most biochemical indicators of biological age were primarily assessed at baseline, limiting our ability to explore time-varying changes due to infrequent follow-up measurements. Sixth, even after adjusting for known potential confounding factors, the possibility of residual and unmeasured confounding factors may still exist. Seventh, while the use of complete-case analysis was necessary to ensure the robustness and consistency of our biological age estimations, it carries a potential risk of selection bias if biomarker data are not missing completely at random. However, prior research in the UK Biobank indicates that this assumption is largely valid for these biomarkers[67].

As summarized in this research, we found that accelerated biological ageing was associated with an increased risk of incident AAA in a large population study. This finding highlights its potential as a focus in risk assessment and for the development of targeted interventions.

## Data availability
The UK Biobank data that support the findings of this study can be accessed by bona fide researchers when applying to access the UK Biobank research resource to conduct health-related research (https://www.ukbiobank.ac.uk/).

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

## Acknowledgements

The authors appreciate the participants in the UK Biobank for their participation and contribution to the research. The study has been conducted using the UK Biobank Resource under application number 100739. This research was supported by the National Natural Science Foundation of China (grant 82470511 and 82070495 to Chen Yao, 82400569 to Kangjie Wang, 82202251 to Xiong Chen, and 81800420 to Kai Huang), the China Postdoctoral Science Foundation (grant 2022M710884 to Xiong Chen), and the Guangdong Basic and Applied Basic Research Foundation of provincial and municipal joint fund (grant 2023A1515111014 to Kangjie Wang). The funders had no role in the design of the study, collection, analysis, interpretation of data, or writing the manuscript.

## Author contributions

C.Y., G.Y., R.S., and K.H. conceived the study and contributed to the interpretation of the results. G.Y., X.C., and K.H. had full access to and verified all the data. C.Y., G.Y., and R.S. did statistical analyses and drafted the manuscript. K.W., Y.S., and X.C. provided critical revisions of the manuscript for important intellectual content over multiple rounds. K.H. attests that all listed authors meet authorship criteria and that no others meeting the criteria have been omitted. All authors critically reviewed the manuscript and approved the decision to submit for publication.

## Competing interests

The authors declare no competing interests.
