## [Transparent Peer Review file · Communications Medicine]

Associations of biological ageing and genetic risk with incident abdominal aortic aneurysm

Corresponding Author: Dr Kai Huang

Version 0:

Reviewer comments:

Reviewer #2

(Remarks to the Author)

Reviewer #3

(Remarks to the Author)

Thank you for the opportunity to review the paper by Yao et al. The paper is well written and touches upon an interesting and relevant question. Unfortunately, the strategy chosen comes with a number of limitations and in this reviewer's opinion the presented data is inconclusive with respect to an association between biological and AAA disease, let alone of a causal relation between biological age and incident AAA.

A critical question is whether age, as stated, is a risk factor for AAA disease, or that incident AAA relies on prolonged exposure to risk factors and thus simply reflects exposure time.

The authors apply KDMAge and PhenoAge as estimates of biological age, since strong associations are present between AAA disease and factors included in these algorithms, any estimates based on these algorithms might be confounded. Along similar lines, to the best of my knowledge the UK has not introduced a screening program for AAA. As such most diagnosed AAA reflect a co-incidental finding of abdominal imaging and as such a bias towards patients with underlying disease.

With respect to the discussion: "given that AAA of any size can rupture" is obviously correct, like the statement that any artery can rupture. Yet, the AAA community considers the rupture risk of aneurysms below 55 mm as negligible. What anti-ageing strategies should be prioritized (do they, apart from smoking cessation exist?)

Reviewer #4

(Remarks to the Author)

Many thanks for the opportunity to review your work.

The topic is interesting, the findings were well described and the paper was well written.

Just a few minor comments for the authors to consider to improve the clarity and accuracy of their findings:

Mandatory change:

The authors had used an incorrect equation to estimate the PhenoAge (on Page 3 of the Supplementary materials). The correct formula is: $\text{PhenoAge} = 141.50 + \text{LN}[-0.00553 * \text{LN}(1 - \text{Mortality risk})] / 0.09165$. The authors had used the original formula for which a correction had been made subsequently (see: Liu Z, Kuo PL, Horvath S, Crimmins E, Ferrucci L, Levine M. Correction: A new aging measure captures morbidity and mortality risk across diverse subpopulations from NHANES IV: A cohort study. PLOS Med 2019;16(2): e1002760.) which was used in this reviewer's previous work on a similar topic with the correct formula described in the Supplemental materials of the following paper (DOI: 10.1007/s11739-023-03397-3).

Minor change:

1. Given smoking and hypertension are known risk factors for AAA. It will be informative to report their hazard ratios in Table 2. That is, the full Cox model should be presented and not just the hazard ratios of the aging clocks.
2. "First, it is a prospective study" is not strictly correct; data were prospectively collected but it was never designed for this

purpose prospectively. A retrospective cohort study (of prospectively collected data) is probably more correct.
3. It was surprising that KDMAge was less predictive than PhenoAge given the former has a SBP component. Perhaps, this should be discussed in the discussion.
4. A histogram showing the distribution of the PhenoAge & KDMAge residuals will be better than just mean and SD data as in the table S6.

Version 1:

Reviewer comments:

Reviewer #3

(Remarks to the Author)

The authors more than adequately handled my questions raised.

Reviewer #4

(Remarks to the Author)

Thank you very much for the authors' revision of their manuscript addressing my comments on the previous version of the manuscript. I believe they have adequately addressed the comments and I have no further request on how to further improve this study.

Reviewer's comments:

Reviewer #1: Thank you for the opportunity to review this manuscript by Yao et al. They focused on how accelerated biological ageing contributes to age-related diseases, but not that its specific role in AAA risk remains unclear. They sought to explore the relationships between biological ageing and risk of incident AAA and genetic predisposition to the disease. The manuscript is well written and provides interesting and convincing insights. However, there are a few notes of consideration which I list here.

1. Study population: From the methods I gathered that there was no filtering on ethnicity, in point of fact one of the covariates is 'ethnicity' (page 7, line 162). Yet, in the discussion it is noted that "the majority of our samples consisted of middle-aged or older White adults" (page 14, line 344-345). Please elaborate: - - - Did you or did you not filter on ethnicity in these data? What is the ethnic composition of your selection of samples? To what does this sentence refer to?

Response to comment 1:

We sincerely thank you for this insightful and constructive comment, which helps us clarify an important aspect of our study population. You are correct that we did not apply any ethnicity-based filtering in our data selection. Ethnicity was included as a covariate in our models to account for potential population structure, as stated in the Covariates section. The ethnic composition of our full analytical sample ($N = 350,483$) is as follows (as presented in **Table 1**): White: 331,888 (94.7%), Asian: 6,505 (1.9%), Black: 4,516 (1.3%), Others: 6,017 (1.7%), and Missing: 1,557 (0.4%).

The sentence in the Discussion referring to "the majority of our samples consisting of middle-aged or older White adults" was intended to acknowledge the demographic composition of the UK Biobank—not to imply that any filtering had been applied. We recognize that this phrasing may have been misleading, and we have revised the relevant sentence to clearly state that the limited ethnic diversity reflects an inherent characteristic of the cohort, which may affect the generalizability of our findings to other populations¹.

To improve clarity and contextual coherence, we have revised the sentence to read: "Fourth, our participants were predominantly (~95%) composed of middle-aged or older White adults, which may restrict the generalizability of our findings to more diverse populations⁴³." (page 17, line 417-418). We hope this revision provides improved clarity, and we thank you again for your attentive reading and valuable feedback.

References

1. Jiang M, Tian S, Liu S, et al. Accelerated biological aging elevates the risk of cardiometabolic multimorbidity and mortality. *Nat Cardiovasc Res* 2024;3:332-342. doi: <https://doi.org/10.1038/s44161-024-00438-8>

Partial Table 1 for Reviewer 1. Baseline characteristics of the participants included in the study.

	Total participants (N = 350,483)	Participants without AAA (N = 348,597)	Participants with AAA (N = 1,886)
Age (years)	56.4 (8.1)	56.3 (8.1)	63.0 (5.3)
Female sex, n (%)	189,698 (54.1%)	189,363 (54.3%)	335 (17.8%)
Ethnicity, n (%)			
Asian	6,505 (1.9%)	6,498 (1.9%)	7 (0.4%)
Black	4,516 (1.3%)	4,505 (1.3%)	11 (0.6%)
White	331,888 (94.7%)	330,040 (94.7%)	1,848 (98.0%)
Others	6,017 (1.7%)	6,005 (1.7%)	12 (0.6%)
Missing	1,557 (0.4%)	1,549 (0.4%)	8 (0.4%)

2. In the Methods, page 4, line 103-104 you mentioned “excluded participants without traits demanded by the two biological age measurements (n = 147,247)”. Can you

explain: - - - What traits specifically do you refer to? Why are these traits not present? In your discussion you do not refer to this anymore. Yet, I do believe this may introduce a bias. Do you concur and regardless, please elaborate on this point in the discussion.

Response to comment 2:

We sincerely thank you for this valuable and insightful comment, which has provided an important opportunity to clarify a key methodological aspect of our study and discuss its implications. The phrase "excluded participants without traits demanded by the two biological age measurements" refers to the removal of participants with missing values in any biomarker required to calculate KDMAge or PhenoAge. The specific biomarkers involved (as also detailed in the "**Assessment of biological ageing**" section on page 5-6, line 118-136 and **Supplementary Methods**) include:

(1) For KDMAge: forced expiratory volume in one second, systolic blood pressure, albumin, alkaline phosphatase, blood urea nitrogen, creatinine, C-reactive protein, glycated hemoglobin, and total cholesterol.

(2) For PhenoAge: albumin, alkaline phosphatase, creatinine, C-reactive protein, glucose, lymphocyte percentage, mean cell volume, red cell distribution width, and white blood cell count.

To improve clarity, we have revised the original sentence in the Methods section to now read: "In this study, we excluded participants with missing values in any biomarker required for the calculation of the two biological age measures (n = 147,247)" (page 5, line 109-110).

We fully agree with you that the complete-case analysis approach may pose a potential risk of selection bias if data are not missing completely at random. Notably, as noted in prior work using UK Biobank data¹, missingness in these biomarker values appeared to be independent of both the biomarker levels and the participant characteristics, and is primarily attributable to technical issues such as aliquot errors or dilution problems. Accordingly, we proceeded under the assumption that the data were missing completely at random. Moreover, we recognize the potential limitation of complete-case analysis and have added the following statement in the Discussion: "Seventh, while the use of complete-case analysis was necessary to ensure the robustness and consistency of our

biological age estimations, it carries a potential risk of selection bias if biomarker data are not missing completely at random. However, prior research in the UK Biobank indicates that this assumption is largely valid for these biomarkers⁶⁸." (page 17, line 422-426). We are deeply grateful for your insightful comment, which has allowed us to clarify and strengthen the methodological justification in our manuscript.

References

1. Bortz J, Guariglia A, Klaric L, et al. Biological age estimation using circulating blood biomarkers. *Commun Biol* 2023;6:1089. doi: <https://doi.org/10.1038/s42003-023-05456-z>

3. Results: On page 9, line 224-227 you mention the numbers of participants with KDMage and PhenoAge. - Please provide the intersection: the number of patients with both KDMage and PhenoAge available.

Response to comment 3:

We sincerely appreciate your insightful comment. We apologize for any confusion caused by the original wording. To clarify more clearly, during the cohort selection process, we excluded participants with missing values in any biomarker required for the calculation of the two biological age measures (as stated in page 5, line 109-110). Therefore, the final analytic cohort of 350,483 participants all have complete data for both KDMage and PhenoAge measures, as presented in **Table 1**.

In the mentioned sentence (original page 9, line 224-227; now on page 9-10, line 231-234), we were specifically referring to the characteristics of accelerated ageing of biological age measures. As defined on page 5-6, line 131-133: "accelerated biological ageing was characterized by a KDMAge or PhenoAge acceleration value above zero, while non-accelerated biological ageing was identified with the value of zero or less." Accordingly, "171,573 (49.0%) participants of the whole population exhibited accelerated ageing as determined by KDMAge measure, and 162,492 (46.4%) participants were identified as experiencing accelerated ageing according to the PhenoAge metric."

To further clarify the intersection between these groups, we have now added the following sentence: "Among these, 97,933 (27.9%) individuals met the criteria for accelerated ageing on both the KDMAge and PhenoAge measures." (page 10, line 234-235). We sincerely thank you for prompting this clarification and will revise the manuscript accordingly to avoid ambiguity.

Partial Table 1 for Reviewer 1. Baseline characteristics of the participants included in the study.

	Total participants (N = 350,483)	Participants without AAA (N = 348,597)	Participants with AAA (N = 1,886)
Age (years)	56.4 (8.1)	56.3 (8.1)	63.0 (5.3)
...
Biological ages			
KDMAge, years	56.7 (49.1, 62.4)	56.7 (49.1, 62.3)	63.4 (59.4, 66.7)
PhenoAge, years	45.8 (38.1, 52.5)	45.7 (38.0, 52.4)	55.7 (50.3, 60.3)
KDMAge acceleration	-0.1 (-1.6, 1.5)	-0.1 (-1.6, 1.5)	0.3 (-1.3, 2.1)
PhenoAge acceleration	-0.4 (-3.5, 3.0)	-0.4 (-3.5, 3.0)	2.5 (-1.0, 6.1)
KDMAge, non-accelerated ageing	178,910 (51.0%)	178,073 (51.1%)	837 (44.4%)
KDMAge, accelerated ageing	171,573 (49.0%)	170,524 (48.9%)	1,049 (55.6%)
PhenoAge, non-accelerated ageing	187,991 (53.6%)	187,378 (53.8%)	613 (32.5%)
PhenoAge, accelerated ageing	162,492 (46.4%)	161,219 (46.2%)	1,273 (67.5%)

4. On page 10, line 243-245 you mention the association of biological ageing in the

context of BMI. However, this result is counter intuitive to me: “The associations between accelerated biological ageing, as quantified by KDMAge and PhenoAge, and the risk of AAA were stronger in participants with a BMI under 25 kg/m² compared to those with higher BMIs”.

- - Why would the association between KDMAge/PhenoAge and AAA risk be stronger in individuals with a BMI < 25? Wouldn't you expect the association is weaker? Can you explain this? Please elaborate on this point in the discussion.

Response to comment 4:

We sincerely thank you for raising this thoughtful point, which provides a valuable opportunity to further explore and clarify the relationship between biological ageing, BMI, and AAA risk. We agree that the observed stronger association between accelerated biological ageing and AAA risk among participants with BMI ≤ 25 kg/m² appears counterintuitive at first glance.

To address this important comment, we have added the following paragraph to the Discussion section: "The stronger association between accelerated biological ageing and AAA risk observed in participants with BMI ≤ 25 kg/m² may reflect increased susceptibility due to low lean mass, or unintentional weight loss—conditions often associated with frailty and systemic physiological decline, especially while considering aged population^{57,58}. In these individuals, accelerated biological ageing likely coincides with heightened inflammatory states and increased cellular damage, which may synergistically increase AAA susceptibility^{59,60}. In contrast, a moderately elevated BMI may indicate preserved metabolic reserve or muscle mass, potentially mitigating the detrimental effects of biological ageing and thus attenuating the association of biological ageing with AAA risk^{61,62}. This pattern aligns with the “obesity paradox” previously observed in older and frail populations, where mild adiposity has been linked to better survival, possibly owing to greater energy reserves and reduced catabolism⁶³. Furthermore, reverse causality may play a role, as underlying health impairments could simultaneously drive both accelerated biological ageing and weight loss, or the limitations of the BMI index itself, thereby exaggerating this association observed in normal-weight individuals^{64,65}. Future studies would benefit from incorporating

multiple measures of adiposity alongside BMI to better clarify the roles of adiposity and biological ageing in AAA pathogenesis." (page 15-16, line 372-387). We are truly grateful to you for prompting this important clarification, which has significantly strengthened the biological interpretation and clinical relevance of our findings.

5. Discussion: Above I mentioned a few items to add to the discussion. You also refer to causality on page 14 (line 350-351).

- - Are there any genetic factors associated with KDMage/PhenoAge? If so: why not perform a Mendelian randomization study? This would significantly improve the impact of your study.

Response to comment 5:

We sincerely thank you for this insightful suggestion regarding the potential use of Mendelian randomization (MR) to enhance causal inference between biological ageing and AAA. We fully agree that MR represents a powerful approach for evaluating causality using genetic instruments, and we had also carefully considered incorporating this method during the planning stages of our study. Upon thorough evaluation of the currently available genetic resources, we concluded that conducting a methodologically robust MR analysis remains challenging at this point, due to certain constraints pertaining to genetic instruments:

(1) For KDMAge acceleration: To the best of our knowledge, no genome-wide association study (GWAS) of sufficient scale has been conducted to identify robust genetic instruments for this measure. The KDMAge algorithm incorporates a complex combination of biomarkers, which genetic architecture has been not characterized in large public biobanks. The lack of strongly associated and validated genetic variants makes it difficult to construct instrumental variables that satisfy key MR assumptions.

(2) For PhenoAge acceleration: Although a prior available GWAS¹ (PMID: 34038024) was performed in a subset of the UK Biobank (including only 107,460 participants due to its study design), we are concerned that the limited sample size may not fully capture the genetic architecture of PhenoAge acceleration with potentially reduced statistical

power. Given that our own analysis uses the entire UK Biobank cohort with available biomarkers ($N = 350,483$), the currently available genetic summary statistics, derived from a much smaller sample, which may introduce bias and compromise the validity of MR assumptions. In addition, the possibility of horizontal pleiotropy cannot be ruled out, which further complicates causal interpretation.

Therefore, in light of these limitations, especially the absence of well-powered GWAS for KDMAge and PhenoAge accelerations, we concluded that attempting an MR analysis with available data at present would currently face methodological challenges and could compromise the validity of findings.

On the other hand, our study utilized a cohort design based on prospectively collected data from the UK Biobank. This design offers inherent methodological advantages, including reduced recall bias and the ability to establish clear temporality. These strengths allowed us to rigorously adjust for a range of clinical and lifestyle covariates, examine the interaction between accelerated biological ageing and genetic susceptibility to AAA, and ultimately provide insights into the relationship between biological ageing and AAA incidence. And to the best of our knowledge, this work helps address a gap in the existing literature.

We fully agree with you that MR analysis will be an essential direction for future research. As more powerful GWAS for these biological ageing measures (KDMAge/PhenoAge) become available, ideally from consortium-based efforts or the full Biobank, we believe that MR will offer valuable causal insights. We have incorporated a statement in the Discussion to reflect this perspective: "First, our observational study, based on prospectively collected data, reveals an association between accelerated biological ageing and increased AAA risk, but causality is not definitively established. Future Mendelian randomization studies using robust genetic instruments derived from large-scale GWAS data may further elucidate the causal nature of their associations." (page 16-17, line 407-411). Lastly, we sincerely appreciate your thoughtful and valuable comments, which has helped us better articulate the methodological considerations and future research implications.

References

1. Kuo CL, Pilling LC, Liu Z, Atkins JL, Levine ME. Genetic associations for two biological age measures point to distinct aging phenotypes. *Aging Cell* 2021;20:e13376. doi: <https://doi.org/10.1111/ace1.13376>

Reviewer #2: Thank your for the opportunity to review the paper by Yao et al. The paper is well written and touches upon an interesting and relevant question.

1. Unfortunately, the strategy chosen comes with a number of limitations and in this reviewer's opinion the presented data is inconclusive with respect to an association between biological and AAA disease, led alone of a causal relation between biological age and incident AAA.

Response to comment 1:

We sincerely appreciate your thoughtful critique of our work and your emphasis on the challenges inherent in establishing causal relationships from observational data. We agree that observational studies, including ours, cannot definitively prove causality. Nonetheless, we believe the breadth, consistency, and rigor of our analyses provide compelling evidence for a prospective association between accelerated biological ageing and AAA risk, supported by biological plausibility and methodological thoroughness.

Our study leverages the large-scale, prospective design of the UK Biobank, which includes detailed baseline phenotyping and lengthy follow-up (median 13.68 years) in a cohort of 350,483 individuals. We quantified biological age acceleration using two established algorithms, KDMAge and PhenoAge, which integrate multiple clinical biomarkers to capture systemic physiological ageing. Both measures consistently demonstrated dose-dependent associations with incident AAA, independent of chronological age and a comprehensive set of potential confounders, including socioeconomic status, lifestyle factors, cardiometabolic health indicators, and medication use. The hazard ratios of 1.29 (95% CI: 1.17-1.42) for KDMAge acceleration and 1.63 (95% CI: 1.47-1.81) for PhenoAge acceleration indicate that biological ageing contributes to AAA risk above and beyond conventional risk factors and the passage of time alone.

To enhance the robustness of our findings, we conducted multiple sensitivity analyses. A landmark analysis excluding early incident cases helped mitigate concerns of reverse causality, while complete-case analysis and competing risk models produced results

consistent with our primary findings. Furthermore, the observed additive interaction between high genetic risk and PhenoAge acceleration suggests combined effects of genetic predisposition and accelerated ageing, reinforcing the clinical and biological relevance of biological age as a risk marker.

From a biological perspective, accelerated ageing reflects a state of multisystem physiological decline, encompassing like cellular senescence, chronic inflammation, and vascular dysfunction, each of which has been implicated in AAA pathogenesis^{1,2}. The biomarkers included in KDMAge and PhenoAge are likely sensitive to these feedforward pathways, supporting the idea that biological age acceleration serves as an integrative measure of systemic physiological integrity^{2,3}.

In response to this comment, we have supplemented the Discussion to include the following statement: "First, our observational study, based on prospectively collected data, reveals an association between accelerated biological ageing and increased AAA risk, but causality is not definitively established. Future Mendelian randomization studies using robust genetic instruments derived from large-scale GWAS data may further elucidate the causal nature of their associations." (page 16-17, line 407-411).

We hope these revisions enhance the methodological transparency and interpretive rigor of our manuscript. We are truly grateful for your insightful feedback, which has helped us better contextualize our findings and strengthen the overall narrative of the study.

References

1. Hu C, Zhang X, Teng T, Ma ZG, Tang QZ. Cellular Senescence in Cardiovascular Diseases: A Systematic Review. *Aging Dis* 2022;13:103-128. doi: <https://doi.org/10.14336/ad.2021.0927>
2. Talifu Z, Ren Z, Chen C, et al. The Association Between Accelerated Biological Aging and the Physical, Psychological, and Cognitive Multimorbidity and Life Expectancy: Cohort Study. *Aging Cell* 2025:e70142. doi: <https://doi.org/10.1111/accel.70142>
3. Li X, Cao X, Zhang J, et al. Accelerated aging mediates the associations of unhealthy lifestyles with cardiovascular disease, cancer, and mortality. *J Am Geriatr Soc* 2024;72:181-193. doi: <https://doi.org/10.1111/jgs.18611>

2. A critical question is whether age, as stated, is a risk factor for AAA disease, or that incident AAA relies on prolonged exposure to risk factors and thus simply reflects exposure time.

Response to comment 2:

We are deeply grateful to you for raising this fundamental question, which lies at the very heart of our study's objective. We fully agree that prolonged exposure to risk factors over time is a cornerstone of AAA pathogenesis. At the same time, our research was designed to test a more nuanced hypothesis: that individual variance in the trajectories of biological ageing, captured through integrative biomarkers and distinct from mere chronological time, significantly contribute to AAA risk. We appreciate the opportunity to elaborate on our methodological approach and the implications of our findings.

Methodologically, to disentangle the “pace of ageing” from “length of exposure”¹, we focused on biological age acceleration rather than chronological age itself. Biological age was estimated using the KDMAge and PhenoAge algorithms, which integrate multiple clinical biomarkers such as albumin, creatinine, glucose, C-reactive protein, and lymphocyte distribution width. Crucially, we regressed these biological age estimates on chronological age across the entire cohort. The residuals from this model represent biological age acceleration, a metric independent of chronological age. A positive residual indicates an individual whose physiological state is older than the average for their birth cohort. Our Cox proportional hazards models demonstrated that both KDMAge and PhenoAge accelerations significantly predicted AAA risk even after adjusting for chronological age, with hazard ratios of 1.29 (95% CI: 1.17-1.42) and 1.63 (95% CI: 1.47-1.81), respectively. This supports the notion that the rate of physiological decline contributes to AAA risk above and beyond the mere passage of time.

Biologically, we propose that accelerated ageing reflects the cumulative burden of pathological processes central to AAA development. It captures active mechanisms such as cellular senescence, chronic inflammation, and vascular dysfunction, each of

which has been implicated in AAA pathogenesis^{1,2}. For instance, vascular smooth muscle cell senescence, driven by processes such as SIRT1 inhibition and NF- κ B activation, promotes a pro-inflammatory, matrix-degrading microenvironment conducive to aneurysm formation². The biomarkers incorporated in our algorithms are likely sensitive to these feedforward processes; for example, CRP reflects systemic inflammation, and glucose levels indicate metabolic stress^{1,3}. Thus, accelerated biological ageing captures not only the duration but, more importantly, the intensity of exposure over time.

From a clinical and public health perspective, this distinction is highly meaningful. While chronological age is a non-modifiable and relatively coarse predictor of AAA risk, biological age acceleration helps identify individuals who, though possibly chronologically younger, exhibit advanced physiological ageing and are thus at elevated risk. This insight could enable more targeted and efficient screening strategies. Furthermore, biological ageing is potentially modifiable through interventions targeting underlying mechanisms, such as lifestyle changes or emerging senolytic therapies⁴, that may slow ageing processes and reduce AAA risk. Our observation of an additive interaction between genetic risk and PhenoAge acceleration further underscores the potential value of addressing accelerated ageing, especially in genetically susceptible individuals.

In summary, although we fully acknowledge the essential role of time-dependent risk accumulation, our findings highlight that individual variation in the trajectory of biological ageing significantly modulates AAA risk. Biological age acceleration serves as an integrative, clinically informative measure that enhances risk stratification and may open new avenues for further researches.

In response to this comment, we have supplemented the Introduction to include the following statement: "While chronological age serves as a crude proxy for cumulative risk exposure¹⁵, we hypothesize that individual variation in biological ageing trajectories as captured by ageing biomarkers, quantified as biological age acceleration independent of chronological age, may independently contribute to AAA development." (page 4, line 83-86). We hope these clarifications strengthen our

methodological transparency and interpretive rigor, and we thank you again for this insightful comment, which has greatly helped us clarify and strengthen the narrative of our study.

References

1. Talifu Z, Ren Z, Chen C, et al. The Association Between Accelerated Biological Aging and the Physical, Psychological, and Cognitive Multimorbidity and Life Expectancy: Cohort Study. *Aging Cell* 2025:e70142. doi: <https://doi.org/10.1111/accel.70142>
2. Hu C, Zhang X, Teng T, Ma ZG, Tang QZ. Cellular Senescence in Cardiovascular Diseases: A Systematic Review. *Aging Dis* 2022;13:103-128. doi: <https://doi.org/10.14336/ad.2021.0927>
3. Li X, Cao X, Zhang J, et al. Accelerated aging mediates the associations of unhealthy lifestyles with cardiovascular disease, cancer, and mortality. *J Am Geriatr Soc* 2024;72:181-193. doi: <https://doi.org/10.1111/jgs.18611>
4. Furrer R, Handschin C. Biomarkers of aging: from molecules and surrogates to physiology and function. *Physiol Rev* 2025;105:1609-1694. doi: <https://doi.org/10.1152/physrev.00045.2024>

3. The authors apply KDMAge and PhenoAge as estimates of biological age, since strong associations are present between AAA disease and factors included in these algorithms, any estimates based on these algorithms might be confounded.

Response to comment 3:

We sincerely thank you for raising this important point regarding potential confounding in our use of KDMAge and PhenoAge as estimators of biological age, particularly since some of their constituent biomarkers are established risk factors for AAA. We appreciate the opportunity to clarify our methodological approach and the conceptual basis of these measures.

It is true that both KDMAge and PhenoAge incorporate biomarkers such as systolic blood pressure and C-reactive protein, which are linked to AAA. However, these algorithms are designed to capture integrative, systemic ageing processes rather than serving as mere collections of discrete risk factors. They reflect emergent properties

derived from multivariate interactions among biomarkers, providing a holistic measure of biological ageing. Moreover, our analysis primarily used biological age acceleration (defined as the residuals from regressing biological age on chronological age) which inherently adjusts for chronological age and helps isolate ageing-specific physiological deviations. To further address potential confounding, our Cox models were rigorously adjusted for a comprehensive set of lifestyle and clinical covariates, including age, sex, ethnicity, body mass index, smoking status and pack-years, alcohol consumption, physical activity, diet, socioeconomic status (Townsend Deprivation Index), employment, self-reported health, and use of blood pressure, cholesterol-lowering, and insulin medications. Although we did not adjust for individual biomarkers such as SBP or CRP, as they are intrinsic components of the ageing algorithms, the inclusion of these covariates helps account for underlying risk factors that these biomarkers may reflect. The persistence of strong associations between biological age acceleration and AAA incidence after these adjustments suggests that accelerated ageing captures risk beyond conventional factors alone.

Moreover, the conceptual foundation of KDMAge and PhenoAge supports their validity as integrative measures of physiological decline. Both have been widely validated in relation to ageing-related outcomes and mortality across diverse populations¹⁻³. Their association with AAA is likely attributable to related mechanisms of systemic dysregulation, such as chronic inflammation, and vascular dysfunction, that potentially underlie both ageing and AAA pathogenesis^{4,5}. Thus, rather than introducing confounding, the incorporation of AAA-relevant biomarkers underscores that AAA is an age-related disease rooted in multisystem physiological impairment.

In response to this comment, we have supplemented the Discussion to include the following statement: "It should be noted that while KDMAge and PhenoAge incorporate biomarkers such as C-reactive protein and systolic blood pressure, which are established risk factors for AAA^{50,51}, these algorithms are designed to capture integrative, systemic ageing processes rather than functioning merely as collections of discrete risk factors⁵². Further, our use of age acceleration residuals inherently adjusts for chronological age, thereby capturing the variance in biological age attributable to

divergent ageing trajectories⁵³. This approach allows us to test whether the ageing-related physiological dysregulation captured by these algorithms is associated with AAA beyond the contribution of individual biomarkers. Additionally, the association between biological age acceleration and AAA incidence persisted even after rigorous adjustment for a set of lifestyle and clinical covariates. This suggests that the predictive utility of these measures likely stems from their ability to capture physiological dysregulation, including feedforward pathways like vascular dysfunction and chronic inflammation, which underlie both ageing and AAA pathogenesis^{31,54-56}. Further studies are needed to elucidate whether these biomarkers contribute to AAA development through specific ageing pathways." (page 15, line 358-371). We hope these clarifications strengthen our methodological transparency and interpretive rigor, and we truly appreciate your constructive feedback, which has helped improve the manuscript's clarity.

References

1. Wang M, Yan H, Zhang Y, et al. Accelerated biological aging increases the risk of short- and long-term stroke prognosis in patients with ischemic stroke or TIA. *EBioMedicine* 2025;111:105494. doi: <https://doi.org/10.1016/j.ebiom.2024.105494>
2. Xiang H, Huang Y, Zhang Y, et al. Clinical biomarker-based biological ageing and the risk of adverse outcomes in patients with chronic kidney disease. *Age Ageing* 2024;53. doi: <https://doi.org/10.1093/ageing/afae245>
3. Wang S, Prizment A, Moshele P, et al. Aging measures and cancer in the Health and Retirement Study (HRS). *Nat Commun* 2025;16:5916. doi: <https://doi.org/10.1038/s41467-025-60913-z>
4. Ajoalabady A, Pratico D, Vinciguerra M, et al. Inflammaging: mechanisms and role in the cardiac and vasculature. *Trends Endocrinol Metab* 2023;34:373-387. doi: <https://doi.org/10.1016/j.tem.2023.03.005>
5. Zeng Z, Yu C, Chen R, et al. Biological aging and incident cardiovascular diseases in individuals with diabetes: insights from a large prospective cohort study. *Cardiovasc Diabetol* 2025;24:320. doi: <https://doi.org/10.1186/s12933-025-02855-w>

4. Along similar lines, to the best of my knowledge the UK has not introduced a screening program for AAA. As such most diagnosed AAA reflect a co-incidental finding of abdominal imaging and as such a bias towards patients with underlying disease.

Response to comment 4:

We sincerely thank you for raising this important point. We appreciate your insightful comment regarding the potential for selection bias in AAA diagnosis within the UK Biobank, particularly in the context of screening practices. You are entirely correct to highlight that the detection of AAA can often be incidental during abdominal imaging for other conditions, which could introduce a bias toward diagnosing patients with comorbid illnesses. We have supplemented this inherent limitation in our study's discussion by stating: "Third, our diagnosis of outcome events relies on hospital inpatient and death records, possibly missing latent cases exhibiting less pronounced clinical symptoms. Furthermore, despite the existence of the UK's national AAA screening programme⁶⁷, some cases may still arise from incidental detection during other imaging, potentially introducing selection bias towards individuals with higher comorbidity." (page 17, line 412-416). This means our identified cases likely represent more advanced or symptomatic AAAs that came to clinical attention, and we might have missed certain latent cases.

However, we would like to respectfully clarify that a national AAA screening program does exist in UK. The NHS abdominal aortic aneurysm screening programme has invited all men aged 65 for an abdominal ultrasound scan (<https://www.gov.uk/guidance/abdominal-aortic-aneurysm-screening-programme-overview>). The program's explicit goal is the early detection of AAAs to reduce mortality from rupture. The latest annual standards report confirms its ongoing operation (<https://nationalscreening.blog.gov.uk/2024/10/15/nhs-abdominal-aortic-aneurysm-aaa-screening-annual-standards-report-published/>). It should be acknowledged that while this national screening programme is an important source of case identification, incidental diagnoses in clinical practice remain a significant

contributor, as the programme does not cover all individuals in the country.

Our AAA outcome definition, based on ICD-10 and OPCS-4 codes from hospital inpatient and death registry data, is the standard methodology used by large-scale epidemiological studies and previous high-impact publications in the field (e.g., *European Heart Journal*¹, *Circulation*²). This approach ensures the captured events are clinically significant. While this method may not capture every single subclinical aneurysm, it robustly identifies those that have progressed to a stage requiring medical intervention or contributing to mortality, which are the outcomes of greatest clinical and public health interest.

Therefore, while the potential for some detection bias exists, we believe it is unlikely to substantially alter the associations between biological ageing and AAA risk. The large sample size, prospective design of UK Biobank, and consistency of our findings with the established biology of ageing strengthen the validity of our conclusions. Thank you again for this valuable comment, which has allowed us to provide further context for our methodology.

References

1. Ma Y, Li D, Cui F, et al. Air pollutants, genetic susceptibility, and abdominal aortic aneurysm risk: a prospective study. *Eur Heart J* 2024;45:1030-1039. doi: <https://doi.org/10.1093/eurheartj/ehad886>
2. Ho FK, Mark PB, Lees JS, et al. A Proteomics-Based Approach for Prediction of Different Cardiovascular Diseases and Dementia. *Circulation* 2025;151:277-287. doi: <https://doi.org/10.1161/circulationaha.124.070454>

5. With respect to the discussion: "given that AAA of any size can rupture" is obviously correct, like the statement that any artery can rupture. Yet, the AAA community considers the rupture risk of aneurysms below 55 mm as negligible. What anti-ageing strategies should be prioritized (do they, apart from smoking cessation exist?)

Response to comment 5:

We sincerely thank you for raising this important point, which has provided us with an opportunity to clarify and enhance the clinical relevance of our discussion.

(1) We apologize for any lack of clarity in our original statement. You rightly point out that current clinical guidelines regard the rupture risk of small AAAs (<55 mm in men; <50 mm in women) as negligible. We have revised the relevant sentence to better reflect this consensus: "Furthermore, although current guidelines regard the rupture risk of small AAAs (<55 mm in men, <50 mm in women) as negligible, progression into larger aneurysms remains a significant contributor to rupture and even mortality^{46,47}. Given that PhenoAge is intrinsically calibrated to mortality risk, it may be better suited to reflect the systemic deterioration related to AAA development." (page 14, line 344-348).

(2) We also greatly appreciate your question regarding anti-ageing strategies beyond smoking cessation. Indeed, although smoking cessation remains the most well-established intervention for reducing AAA risk and slowing biological ageing, several other evidence-based and emerging approaches show considerable promise^{1,2}, particularly in high-risk individuals.

These include lifestyle interventions such as regular physical activity, which supports vascular health and telomere maintenance, and adherence to a balanced diet that provides all essential macro- and micronutrients within a controlled caloric intake. Reducing consumption of ultra-processed foods is important to lower the risk of obesity, cardiovascular disease, and metabolic disorders, thereby promoting lifelong health maintenance. Additionally, obtaining adequate sleep—generally 7-9 hours per night for young adults and 7-8 hours for older adults, with minimal interruptions, insomnia, or snoring—is important for overall health. Improvements in sleep patterns, including weekend catch-up sleep, have been associated with reduced mortality risk. Furthermore, emerging pharmacological strategies with potential senolytic properties, such as metformin, rapamycin, and novel senolytics, are currently under investigation in ageing-related research. Although these agents have not yet been specifically validated for AAA prevention, they represent compelling candidates for future tailored interventions.

We fully agree that further research is essential to evaluate the efficacy of these

strategies in AAA prevention. In response to your comment, we have updated the Discussion as follows: "While smoking cessation remains the cornerstone of AAA risk reduction, future efforts could benefit from incorporating a broader range of evidence-based anti-ageing strategies. These may include regular physical activity, dietary modifications, restorative sleep, and potential pharmacological agents targeting fundamental ageing processes^{48,49}. Although these approaches are not yet specifically approved for AAA prevention, they show promise in mitigating ageing and warrant further exploration in populations with accelerated ageing." (page 14, line 351-357). We truly thank you for these insightful and valuable comments, which have significantly enhanced the clinical and scientific clarity of our manuscript.

References

1. Furrer R, Handschin C. Biomarkers of aging: from molecules and surrogates to physiology and function. *Physiol Rev* 2025;105:1609-1694. doi: <https://doi.org/10.1152/physrev.00045.2024>
2. Li X, Cao X, Zhang J, et al. Accelerated aging mediates the associations of unhealthy lifestyles with cardiovascular disease, cancer, and mortality. *J Am Geriatr Soc* 2024;72:181-193. doi: <https://doi.org/10.1111/jgs.18611>

Reviewer #3: Many thanks for the opportunity to review your work. The topic is interesting, the findings were well described and the paper was well written. Just a few minor comments for the authors to consider to improve the clarity and accuracy of their findings:

Mandatory change:

The authors had used an incorrect equation to estimate the PhenoAge (on Page 3 of the Supplementary materials). The correct formula is: $\text{PhenoAge} = 141.50 + \text{LN}[-0.00553 * \text{LN}(1 - \text{Mortality risk})] / 0.09165$. The authors had used the original formula for which a correction had been made subsequently (see: Liu Z, Kuo PL, Horvath S, Crimmins E, Ferrucci L, Levine M. Correction: A new aging measure captures morbidity and mortality risk across diverse subpopulations from NHANES IV: A cohort study. *PLOS Med* 2019;16(2): e1002760.) which was used in this reviewer's previous work on a similar topic with the correct formula described in the Supplemental materials of the following paper (DOI: 10.1007/s11739-023-03397-3).

Response to mandatory change:

We sincerely thank you for this critical and highly valuable comment. We deeply appreciate your expertise and careful attention to this important methodological detail. You are correct that we used an outdated equation for PhenoAge estimation in the Supplementary Material. We are grateful for your guidance in directing us to the corrected formula published in *PLOS Medicine* (Liu et al., 2019) as well as the clear methodological example provided in the reviewer's previous work (DOI: 10.1007/s11739-023-03397-3), from which we have learned a great deal.

We have now revised the manuscript to implement the correct formula in all relevant sections. Importantly, we would like to clarify that all our actual analyses were performed using the validated R package *BioAge*

(<https://github.com/dayoonkwon/BioAge>), which implements the corrected formula.

Therefore, this textual error does not affect any of the reported results or conclusions.

The specific revisions we have made are as follows:

(1) In the main text, we have updated the description of PhenoAge with above

appropriate citations: " PhenoAge was derived with nine multisystem blood chemistry biomarkers including four coincide with KDMAge (albumin, alkaline phosphatase, creatinine, C-reactive protein, glucose, lymphocyte proportion, mean cell volume, red cell distribution width, and white blood cell count)^{18,19.}" (page 5, line 123-126)

(2) In the Supplementary Material (page 3-4), we have replaced the equation with the correct formula and provided above appropriate citations in the References:

The formula^{3,4} is:

$$\text{PhenoAge} = 141.50 + \frac{\ln[-0.00553 \times \ln(1 - \text{mortality risk})]}{0.09165}$$

$$\text{mortality risk} = 1 - \exp\left(\frac{-1.51714 \times \exp(xb)}{0.0076927}\right)$$

$xb = -19.907 - 0.0336 \times \text{albumin} + 0.0095 \times \text{creatinine} + 0.1953 \times \text{glucose} + 0.0954 \times \ln(\text{C-reactive protein}) - 0.0120 \times \text{lymphocyte proportion} + 0.0268 \times \text{mean cell volume} + 0.3306 \times \text{red blood cell distribution width} + 0.00188 \times \text{alkaline phosphatase} + 0.0554 \times \text{white blood cell count} + 0.0804 \times \text{chronological age}.$

We are truly grateful to you for identifying this error and providing the necessary references to correct it. This comment has significantly improved the methodological accuracy of our manuscript.

Minor change:

1. Given smoking and hypertension are known risk factors for AAA. It will be informative to report their hazard ratios in Table 2. That is, the full Cox model should be presented and not just the hazard ratios of the aging clocks.

Response to comment 1:

We sincerely thank you for this insightful comment. We fully agree that smoking and hypertension are well-established risk factors for AAA, and appreciate the suggestion to enhance the transparency of our Cox model results. In response, we have now included the hazard ratios for all covariates, including smoking status and hypertension, in the full Cox regression model. Due to space limitations in the main text, these

comprehensive results are presented in Table S7 of the Supplementary Materials and we hope that this operation will meet with your approval.

Additionally, we have added the following clarification in the Results section: "In our analyses, we found that for every 2.386-unit increase in KDMAge acceleration and 5.045-unit increase in PhenoAge acceleration, the risk of incident AAA increased by 6% (Table 2, see also Table S7 for full Cox model estimates)." (page 10, line 239-242). We hope this addition provides a more complete picture of the modelling approach and outcomes. We are very grateful to you for this valuable suggestion, which has strengthened the clarity and reproducibility of our results.

Owing to length limitations, please refer to the accompanying Supplementary Material file for complete Table S7.

Partial Table S7 for Reviewer 3. Hazard ratios for incident abdominal aortic aneurysm risk from full Cox models including biological age acceleration and covariates.

Variables	Model 1 ^a		Model 2 ^b		Model 3 ^c	
	Hazard ratio (95% CI)	P value	Hazard ratio (95% CI)	P value	Hazard ratio (95% CI)	P value
KDMAge acceleration						
Quartile 1	Ref.	-	-	-	-	-
Quartile 2	1.233 (1.070, 1.420)	0.004	-	-	-	-
Quartile 3	1.300 (1.129, 1.496)	2.63e-4	-	-	-	-
Quartile 4	1.60 (1.39, 1.83)	1.65e-11	-	-	-	-
Continuous, per SD increase	-	-	1.065 (1.044, 1.086)	2.56e-10	-	-
Non-accelerated ageing ^d	-	-	-	-	Ref.	-
Accelerated ageing ^e	-	-	-	-	1.291 (1.171, 1.423)	2.80e-07
...
Current or former smoker	1.956 (1.726, 2.216)	< 2e-16	1.955 (1.726, 2.215)	< 2e-16	1.955 (1.726, 2.215)	< 2e-16
Smoking pack-years	1.015 (1.014, 1.017)	< 2e-16	1.015 (1.014, 1.017)	< 2e-16	1.016 (1.014, 1.017)	< 2e-16
Blood pressure	1.286 (1.151, 1.423)	8.48e-06	1.286 (1.151, 1.423)	8.75e-06	1.296 (1.160, 1.423)	4.51e-06

medication	1.436)		1.436)		1.448)	
PhenoAge acceleration						
Quartile 1	Ref.		-	-	-	-
Quartile 2	1.322 (1.107, 1.579)	0.002	-	-	-	-
Quartile 3	1.560 (1.316, 1.850)	3.22e-07	-	-	-	-
Quartile 4	2.384 (2.023, 2.809)	< 2e-16	-	-	-	-
Continuous, per SD increase	-	-	1.064 (1.054, 1.074)	< 2e-16	-	-
Non-accelerated ageing ^d	-	-	-	-	Ref.	-
Accelerated ageing ^e	-	-	-	-	1.631 (1.470, 1.809)	< 2e-16
...
Current or former smoker	1.905 (1.681, 2.159)	< 2e-16	1.904 (1.680, 2.158)	< 2e-16	1.921 (1.695, 2.176)	< 2e-16
Smoking pack-years	1.015 (1.013, 1.017)	< 2e-16	1.015 (1.013, 1.017)	< 2e-16	1.015 (1.014, 1.017)	< 2e-16
Blood pressure medication	1.272 (1.138, 1.421)	2.11e-05	1.263 (1.131, 1.411)	3.59e-05	1.289 (1.154, 1.440)	7.23e-06

Model was adjusted for age, sex, ethnicity, body mass index, education, employment, Townsend Deprivation Index, smoking status, pack-years of smoking, alcohol consumption frequency, healthy diet score, physical activity, self-reported health, blood pressure medication, insulin medication, and cholesterol lowering medication. CI indicates confidence interval; KDMAge, biological age calculated by the Klemmera-Doubal method; PhenoAge, phenotypic age; SD, standard deviation.

^a Model 1 was developed using biological age acceleration categories from quartile 1 to quartile 4.

^b Model 2 was developed using continuous biological age acceleration (per SD increase).

^c Model 3 was developed using biological age acceleration categorized as accelerated vs. non-accelerated ageing.

^d Non-accelerated ageing indicates that the residual of the regression of KDMAge or PhenoAge based on chronological age is ≤ 0 .

^e Accelerated ageing indicates that the residual of the regression of KDMAge or PhenoAge based on chronological age is > 0 .

2. "First, it is a prospective study" is not strictly correct; data were prospectively

collected but it was never designed for this purpose prospectively. A retrospective cohort study (of prospectively collected data) is probably more correct.

Response to comment 2:

We sincerely thank you for this insightful and precise comment. We fully agree that describing our study as "prospective" is not strictly correct, as the data, while collected prospectively within the UK Biobank, were not originally gathered for the specific purpose of this analysis.

In response to this valuable feedback, we have carefully revised the relevant descriptions across the entire manuscript to more accurately reflect the study design. The phrase here has also been updated to: "First, it is a retrospective cohort study of prospectively collected data from the UK Biobank" (page 16, line 400-401). We hope this amendment more appropriately characterizes the nature of our investigation. We are very grateful to you for highlighting this important distinction, which has undoubtedly improved the methodological accuracy of our manuscript.

3. It was surprising that KDMAge was less predictive than PhenoAge given the former has a SBP component. Perhaps, this should be discussed in the discussion.

Response to comment 3:

We sincerely thank you for this thoughtful and insightful comment. We agree that it was surprising that KDMAge—which incorporates SBP, a known risk factor for AAA—showed lower predictive performance than PhenoAge.

In response to this valuable feedback, we have added the following paragraph to the Discussion section: "Although systolic blood pressure, a well-established risk factor for AAA⁴², is incorporated into KDMAge, PhenoAge acceleration demonstrated better risk stratification performance. This discrepancy could be attributed to PhenoAge being designed to predict all-cause mortality using a broader panel of multisystem biomarkers alongside chronological age, enabling it to better capture a range of physiological decline related to inflammation, immune function, and metabolic dysregulation^{43,44}. These processes are collectively relevant to the multisystem impairments involved in

AAA pathogenesis^{11,45}. Furthermore, although current guidelines regard the rupture risk of small AAAs (<55 mm in men, <50 mm in women) as negligible, progression into larger aneurysms remains a significant contributor to rupture and even mortality^{46,47}. Given that PhenoAge is intrinsically calibrated to mortality risk, it may be better suited to reflect the systemic deterioration related to AAA development." (page 14, line 337-348). We hope this addition offers a plausible explanation for the observed difference between the two biological ageing measures. We are very grateful to you for raising this point, which has helped us enrich the discussion and strengthen the interpretive depth of our manuscript.

4. A histogram showing the distribution of the PhenoAge & KDMAge residuals will be better than just mean and SD data as in the table S6.

Response to comment 4:

We sincerely thank you for this valuable suggestion. We agree that a histogram would provide a more intuitive visualization of the distribution of biological age accelerations than summary statistics alone.

In accordance with your comment, we have now created **Figure S1**, which shows the histograms of both PhenoAge and KDMAge acceleration. We have also added the following sentence in the Results section: "Biological age acceleration distributions are shown in Table S6 and Figure S1." (page 10, line 235-236). We hope this addition enhances the clarity and interpretability of our results. We are very grateful to you for this helpful suggestion, which has strengthened the graphical presentation of our study.

Figure S1 for Reviewer 3. Histograms of KDMAge and PhenoAge acceleration in the study.

Histograms of (A) KDMAge acceleration and (B) PhenoAge acceleration in the study ($n = 350,483$). KDMAge, biological age calculated by the Klemera-Doubal method; PhenoAge, phenotypic age.

Reviewer's comments:

Reviewer #3: The authors more than adequately handled my questions raised.

Response:

We sincerely thank you for your positive feedback on our study. We are pleased to know that the reviewer has no further questions, and we deeply appreciate the time and expertise devoted to reviewing our manuscript. Should any further clarifications or refinements be needed, we would be happy to provide them. Thank you once again for your valuable comments.

Reviewer #4: Thank you very much for the authors' revision of their manuscript addressing my comments on the previous version of the manuscript. I believe they have adequately addressed the comments and I have no further request on how to further improve this study.

Response:

We sincerely appreciate your kind acknowledgment of our revisions and are pleased to know that our responses have adequately addressed the comments. Thank you once again for your constructive comments and valuable feedback, which have undoubtedly helped improve the quality of our manuscript. We are grateful for the time and expertise you dedicated to reviewing our work.